# Enhancer Clusters Drive Type I Interferon-Induced TRAIL Overexpression in Cancer, and Its Intracellular Protein Accumulation Fails to Induce Apoptosis

**DOI:** 10.3390/cancers15030967

**Published:** 2023-02-03

**Authors:** Carolina Di Benedetto, Taimoor Khan, Santiago Serrano-Saenz, Anthony Rodriguez-Lemus, Chananat Klomsiri, Tim-Mathis Beutel, Alysia Thach, Henning Walczak, Paola Betancur

**Affiliations:** 1Department of Radiation Oncology, University of California, San Francisco (UCSF), San Francisco, CA 94143, USA; 2CECAD Cluster of Excellence, University of Cologne, 50931 Cologne, Germany; 3Center for Biochemistry, Medical Faculty, University of Cologne, 50931 Cologne, Germany; 4Centre for Cell Death, Cancer, and Inflammation (CCCI), UCL Cancer Institute, University College London, London WC1E 6DD, UK

**Keywords:** TRAIL, enhancer clusters, interferon, intracellular protein accumulation, apoptosis, breast cancer, lung cancer

## Abstract

**Simple Summary:**

Upon interferon stimulation, cancer cells upregulate the pro-apoptotic cytokine TRAIL, but the mechanism of this upregulation remains unresolved. By examining the genomic regulatory landscape of TRAIL in cancer cells, we found that TRAIL is associated with large, densely clustered regulatory enhancers and that these potent enhancer clusters mediate the interferon-driven upregulation of TRAIL in cancer cells. At the protein level, we, surprisingly, found that this interferon-induced TRAIL is not secreted. Instead, it accumulates intracellularly and is thus not capable of inducing apoptosis in cancer cells. Thus, we identified a novel gene regulatory mechanism involving enhancer clusters that explains the high levels of TRAIL expression often encountered in cancer cells. Our results also suggest that the accumulation of interferon-induced TRAIL may be a factor contributing to apoptosis resistance in certain cancer types, a new role that has not been reported for TRAIL before, which deserves further investigation.

**Abstract:**

Tumor necrosis factor-related apoptosis-inducing ligand (TRAIL) is a cytokine produced and secreted by immune cells in response to an infection, often in response to interferon (IFN) stimulation. In cancer, it has also been shown that IFN stimulates the production of TRAIL, and it has been proposed that this TRAIL can induce apoptosis in an autocrine or paracrine manner in different cancer cells. Yet, the mechanism mediating TRAIL upregulation and the implications of TRAIL as an apoptotic molecule in cancer cells are still poorly understood. We show here that in certain cancer cells, TRAIL is upregulated by enhancer clusters, potent genomic regulatory regions containing densely packed enhancers that have combinatorial and additive activity and that are usually found to be associated with cancer-promoting genes. Moreover, we found that TRAIL upregulation by IFNα is mediated by these enhancer clusters in breast and lung cancer cells. Surprisingly, IFNα stimulation leads to the intracellular accumulation of TRAIL protein in these cancer cells. Consequently, this TRAIL is not capable of inducing apoptosis. Our study provides novel insights into the mechanism behind the interferon-mediated upregulation of TRAIL and its protein accumulation in cancer cells. Further investigation is required to understand the role of intracellular TRAIL or depict the mechanisms mediating its apoptosis impairment in cancer cells.

## 1. Introduction

During an immune response, different types of immune cells have been shown to produce the death ligand TRAIL, and this TRAIL has in turn been shown to be capable of inducing apoptosis in TRAIL-sensitive target cells [1]. For instance, activated cytotoxic T lymphocytes and natural killer cells express TRAIL on their surface, and upon interaction with its cognate death receptors, TRAIL-R1 (DR4) and TRAIL-R2 (DR5), on the target cells, it can kill these cells [2,3,4]. Therefore, TRAIL plays a critical role in viral infections and immune surveillance of tumors [5,6,7]. To date, it is known that IFN stimulates the production of TRAIL in immune cells, and more intriguingly, IFN has also been described to upregulate TRAIL in cancer cells [8,9]. For instance, TRAIL upregulation by IFNα, a member of the type 1 Interferon family, has been linked with increased apoptosis induction in an autocrine manner but only in a fraction of cancer cell lines [8,10]. However, the regulatory mechanisms by which TRAIL is upregulated in cancer cells upon IFNα stimulation and the implications of this upregulation in apoptosis induction are still unclear. 

The chromatin landscape harbors epigenetically marked cis-regulatory genomic elements and their associated trans-acting factors. These molecular complexes are required for the precise control of transcriptional gene expression [11,12,13]. Cis-regulatory elements can be found isolated and dispersed throughout the DNA or in clusters. Enhancer clusters, or stretch enhancers, are long stretches of DNA containing highly packed functional enhancers that, in a coordinated fashion, modulate the expression of genes of cell identity as well as oncogenes [14,15]. These hyperactive cis-regulatory elements are enriched in acetylation at histone 3 lysine 27 (H3K27ac), a hallmark of open chromatin that allows DNA to be accessible to the transcriptional machinery [16]. Bromodomain-containing protein 4 (BRD4), a member of the Bromodomain and Extraterminal (BET) family, works as a bridge between hyper-acetylated chromatin regions, thus enhancer clusters, and the transcriptional machinery for the regulation of gene expression [17]. Thus, the transcriptional enhancing function of enhancer clusters can be blocked using inhibitors of BRD4 [18], which have been extensively used to study and address the role of these long, accessible, and highly acetylated regulatory regions in the transcriptional regulation of genes of tumor progression or resistance to cancer therapies [18,19,20,21,22]. 

In this work, we reveal a novel regulatory mechanism involving enhancer clusters in the upregulation of TRAIL upon IFNα isoform 2 (IFNα2) stimulation of breast and lung cancer cells. Our computational findings indicate a possible association of enhancer clusters and TRAIL upregulation in certain cancer cell lines. This was experimentally validated using BET inhibitors, which pharmacologically disrupted these long enhancer-enriched regions and resulted in the reduction of IFNα-induced TRAIL expression, specifically in the cancer cell lines in which TRAIL is associated with enhancer clusters. Surprisingly, upon IFNα stimulation, we also observed intracellular accumulation of TRAIL protein in epithelial cancer cells, as well as absence of TRAIL secretion, independently of TRAIL association with enhancer clusters. Moreover, TRAIL upregulation by IFNα failed to induce apoptosis in most breast and lung cancer cells analyzed. Overall, our study provides novel insight by demonstrating enhancer clusters as a previously unrecognized regulatory mechanism for TRAIL gene expression. In addition, we surprisingly found that intracellularly accumulated TRAIL, induced in cancer cells by stimulation with IFNα, was not capable of inducing apoptosis in any of the cancer cell lines we studied.

## 2. Materials and Methods

### 2.1. In Silico Identification of Enhancer-Enriched Regions

Publicly available chromatin immunoprecipitation followed by sequencing (ChIP–seq) data targeting H3K27ac for a panel of human cancer cell types were downloaded from the Gene Expression Omnibus (GEO) database “http://www.ncbi.nlm.nih.gov/geo/” (accessed on 15 January 2022) or from Encyclopedia of DNA Elements (ENCODE) [23]. Wiggle or BigWig files for H3K27ac signal, aligned to the hg19 human reference genome, were visualized using University of California, Santa Cruz (UCSC), Genome Browser [24]. Accession numbers for the datasets used are listed in Appendix A. The identification of distal regulatory elements upstream and downstream of TRAIL was performed using the “DNase I Hypersensitivity Signal Colored by Similarity from ENCODE” track on UCSC Genome Browser. Only DNase I hypersensitive distal regions found in more than 20 cell types on the “DNase I Hypersensitivity Peak Clusters from ENCODE (95 cell types)” track were considered for the analysis.

### 2.2. Analyses of Cancer Datasets 

Assay for Transposase-Accessible Chromatin with high-throughput sequencing (ATAC-seq) data for the identified DNase I hypersensitive regions, together with TRAIL RNA sequencing (RNA-seq) data available for 404 pan-cancer tumors on The Cancer Genome Atlas (TCGA) dataset [25] were downloaded from UCSC XenaBrowser Platform [26]. Only DNase I hypersensitive DNA regions for which ATAC-seq information was available on TCGA were considered for further analysis. Genetic alterations in TRAIL receptors in breast and lung cancer cell lines on The Cell Line Encyclopedia [27] were analyzed using cBioPortal [28].

### 2.3. Cell Culture 

MCF7, BT549, HCC1954, MDAMB453, THP-1, HeLa and HepG2 cell lines were obtained from ATCC. A549 and T47D were kindly provided by Dr. Mary Helen Barcellos-Hoff at UCSF. MDAMB231 was kindly provided by Dr. Jean-Philippe Coppe at UCSF. MCF7, BT549, HCC1954, MDAMB453, T47D and THP-1 cells were cultured in RPMI medium (Gibco, New York, NY, USA). MDAMB231, HeLa and HepG2 cells were cultured in DMEM medium (Gibco). A549 cells were cultured in F-12K Nutrient Mixture medium (Gibco). Media were supplemented with 10% heat-inactivated FBS (Atlanta Biologicals, Flowery Branch, GA, USA), 100 units/mL penicillin and 100 µg/mL streptomycin (Gibco). In addition, culture media for BT549 and T47D cells were supplemented with 0.01 mg/mL insulin (Sigma, St. Louis, MO, USA), whereas THP-1 culture medium was supplemented with 0.05 mM 2-mercaptoethanol (Gibco). Cells were grown at 37 °C in a humified atmosphere at 5% CO_2_.

### 2.4. Cytokines and Drugs 

For cytokine treatment, Recombinant Human IFNα2 (Biolegend, San Diego, CA, USA), Recombinant Human soluble TRAIL (PeproTech, Cranbury, NJ, USA), Recombinant Human IFNγ (Gibco), Recombinant Human Tumor Necrosis Factor α (TNFα, PeproTech) or Recombinant Human Interleukin-6 (IL-6, PeproTech) dissolved in PBS was added to the cell media at the concentrations indicated in the figures. 

For BET inhibition, cells were treated with 1 µM JQ1 (Sigma) or 1 µM I-BET151 (GSK1210151A; Selleckchem). For the inhibition of pan-caspases, Z-VAD-FMK (A1902-25; Apexbio) was used at a final concentration of 20 µM. For sensitization to TRAIL-induced cell death, cells were treated with 1µM BV6 (S7597; Selleckchem). All drugs were dissolved in DMSO; DMSO was used as vehicle control.

### 2.5. RNA Extraction, cDNA Synthesis and qPCR 

Total RNA was extracted using RNeasy Plus kits (Qiagen, Maryland, USA). cDNA was reverse-transcribed using SuperScript III First-Strand Synthesis SuperMix (Invitrogen, Waltham, MA, USA) and then amplified with QuantStudio 5 Real Time PCR System (Applied Biosystems, Bedford, MA, USA). Specific primers designed to amplify the gene of interest were combined with cDNA and Platinum SYBR Green qPCR SuperMix-UDG (Invitrogen) following the manufacturer’s instructions. qPCR was carried out using the following method: an initial incubation at 50 °C for 2 min followed by incubation at 95 °C for 2 min; 40 cycles at 95 °C for 10 s followed by incubation at 60 °C for 30 s; and a final step for melting curve generation. Results were analyzed using the comparative Ct method [29]. Values were normalized to β-Actin expression. The primers used in this study were:

TRAIL: 

GGGACCAGAGGAAGAAGCAAC, TCATTCTTGGAGTTTGGAGAAGACA

β-Actin: 

TCCCTGGAGAAGAGCTACG, GTAGTTTCGTGGATGCCACA

### 2.6. ELISA 

The quantitative detection of TRAIL was performed in undiluted cell supernatants in duplicates using TRAIL Human ELISA Kit (BMS2004; Invitrogen), following the manufacturer’s instructions. Absorbance was read at 450 nm as the primary wavelength and at 620 nm for reference wavelength on a BioTek plate reader. A standard curve with known TRAIL concentrations was created in each experiment by plotting the mean absorbance for each standard concentration on the ordinate against human TRAIL concentration on the abscissa. TRAIL concentration for each sample was determined by extrapolating the corresponding absorbance values in the standard curve.

### 2.7. Immunofluorescence 

Cells were seeded in 8-well chamber slides (Nunc Lab-Tek Chamber Slide System) and cultured overnight. Following the treatments and incubation times indicated in the figures, cells were rinsed with PBS and fixed with 4% paraformaldehyde for 15 min at room temperature. Cells were washed 2 times with PBS and then permeabilized for 5 min with 0.1% Triton-X100 in PBS. After having been washed 3 times with PBS, samples were blocked with 0.5% casein in PBS for 1 h, then incubated overnight at 4 °C with anti-TRAIL antibody (ATLAS ANTIBODIES; catalog number HPA054938) and diluted to 1:100 in blocking solution. After washing with PBS, cells were stained with a secondary antibody labeled with Alexa Fluor 488 (Abcam; catalog number ab150077) diluted to 1:500; this staining was carried out for 1 h at room temperature. After washing with PBS, DAPI (D1306; Thermo Fisher Scientific, Waltham, MA, USA) was used as nuclear counterstaining. After PBS washes, the samples were mounted overnight with Prolong Glass Antifade mountant (P36982; Thermo Fischer Scientific) before imaging. Each slide was imaged with a 20× Zeiss Plan-Apochromat objective with 0.95 numerical aperture on a Zeiss Axio Observer epifluorescent microscope equipped with a CCD Hamamatsu Photonics monochrome camera with a 1392 × 1040 pixel size at 12 bits per pixel depth. All images were assembled as false-color images using ZEN 2.6-blue edition imaging software (Zeiss, Dublin, CA, USA). Three to five random images were taken for each sample. Mean intensity for TRAIL staining and nuclei counts in each image was measured using an in-house macro with ImageJ software [30]. TRAIL mean intensity was normalized using nuclei counts. 

### 2.8. Flow Cytometry 

For the surface staining of TRAIL on fresh cells, cells were collected, washed in FACS buffer (2% FBS in PBS) and kept at 4 °C for the rest of the protocol. In THP-1 cells, Fc receptors were blocked for 10 min with Human TruStain FcX (Biolegend; catalogue number 422301) diluted to 1:100 in FACS buffer. All the cell lines included in this study were stained for 1 h with APC anti-human CD253 (TRAIL) Antibody (Biolegend; catalogue number 308209). Antibody was used at 1:100 dilution in FACS buffer. After washing, cells were stained for 20 min with LIVE/DEAD Fixable Violet Dead (Life Technologies, Carlsbad, CA, USA) at 1:7500 dilution in PBS. After washing, TRAIL-positive events were quantified using a Northern Light cytometer (Cytek, Fremont, CA, USA). Data were analyzed with SpectroFlo software (Cytek).

For total TRAIL quantification, staining was carried out in fixed and permeabilized cells. First, cells were collected and washed in PBS. Washed cells were stained with LIVE/DEAD Fixable Violet Dead as above, fixed in FluoroFix buffer (Biolegend; catalogue number 422101) for 1 h at room temperature and permeabilized using Intracellular Staining Permeabilization Wash Buffer (Biolegend; catalogue number 421002) following the manufacturer’s instructions. After the blockade of Fc receptors (THP-1 cells only), cells were stained with Polyclonal Antibody against Human TNFSF10 (Atlas Antibodies; catalogue number HPA054938) diluted to 1:100 in Intracellular Staining Permeabilization Wash Buffer followed by staining with Alexa Fluor 488 secondary antibody (Abcam; catalog number ab150077) diluted to 1:1000 in Intracellular Staining Permeabilization Wash Buffer. All antibodies were incubated for 30 min at room temperature. After washing, TRAIL-positive events were quantified using a Northern Light cytometer (Cytek). Data were analyzed with SpectroFlo software (Cytek).

For apoptosis assays, cells were collected, washed with HBSS, and stained for 30 min with APC Annexin V (Biolegend; catalogue number 640919) in Annexin V binding buffer (Biolegend) at room temperature. After washing, Annexin V-positive events were quantified using a Northern Light cytometer (Cytek). Data were analyzed with SpectroFlo software (Cytek).

### 2.9. Cytotoxicity Assay 

Cellular cytotoxicity was measured by detecting the release of the cytosolic enzyme lactate dehydrogenase (LDH) into the cell culture medium upon damage of the cell membrane using CyQUANT LDH Cytotoxicity Assay Kit (Invitrogen), following the manufacturer’s instructions. Briefly, after cytokine treatment for the indicated times, cell culture media were centrifuged to eliminate dead cells and debris and were transferred in triplicates to a 96-well plate. Following 30 min incubation with substrate mix, the reaction was stopped, and absorbance was read at 490 nm and at 680 nm (for background subtraction) on a BioTek plate reader. 

### 2.10. Real-Time Cell Death Assay 

Cells were plated in 96-well plates (at 8000 cells/well) and cultured overnight. Cells were treated as follows: (A) In experiments with conditioned media, HeLa cells were incubated with medium obtained from THP-1 or A549 cells pre-treated for 48 h with IFNα2 or control. The conditioned media were supplemented with 1 μM BV6 to increase HeLa cell sensitivity to cell death. (B) To analyze cell death upon cytokine treatment, A549 cells were incubated with IFNα or TRAIL at the concentrations shown in the figure. Here, all the conditions were also analyzed in cells pre-treated for 1 h with either 20 µM Z-VAD-FMK (for pan-caspase inhibition) or vehicle control. In every case, 1 μg/mL propidium iodide (PI) (R37169; Thermo Fisher Scientific) was added to quantify cell death. Cell death was monitored for 24 h in real time using Incucyte S3 (Incucyte S3 live-cell 7 analysis system; Sartorius) following the manufacturer’s protocol. Four images per well were analyzed. Cell death is represented as the percentage of PI-positive cells in a cell-by-cell analysis.

### 2.11. MTT Assay 

Cells were seeded in 96-well plates (at 7000 cells/well) and cultured overnight. Then, cells were exposed to drug treatments for the indicated times. MTT (Sigma) was added following the manufacturer’s instructions. After 4 h of incubation at 37 °C, formazan crystals were dissolved in 100% DMSO, and absorbance was read at 570 nm on a BioTek plate reader.

### 2.12. Statistical Analysis 

All experiments were performed in triplicates unless otherwise indicated. Results were plotted as mean values +/− standard deviations using GraphPad Prism7 and were statistically analyzed using two-tailed unpaired *t*-test or ANOVA followed by Dunnett’s post-test, as appropriate. *p*-values less than 0.05 were considered statistically significant.

### 2.13. Data Availability

The accession numbers of the publicly available H3K27ac ChIP-Seq datasets analyzed in this study are listed in Appendix A.

## 3. Results

### 3.1. IFNα Upregulates TRAIL Gene Expression and Protein Levels across Different Cancer Types

It was previously shown that the IFN stimulation of different immune cells can induce the expression of TRAIL at both the mRNA and protein levels [31]. To confirm this, we stimulated the THP-1 monocyte cell line with different cytokines, including type I and type II IFNs. We noticed that when THP-1 cells were treated with IFNα (isoform 2), there was a greater than 10-fold increase in TRAIL transcript at 6 h (less at 24 h) when compared with treatment with TNFα, IFNγ or IL-6 (Figure 1A). As expected, we also observed by means of flow cytometry an increase in TRAIL protein expression on the cell surface of THP-1 cells after 24 h of treatment with IFNα (Figure 1B). It was also previously reported that TRAIL is upregulated by IFNα in cancer cells [8]. To confirm this, we performed IFNα stimulation in a panel of cancer lines and assayed TRAIL transcript levels using qPCR. Indeed, in all the cancer lines stimulated with IFNα, TRAIL was significantly upregulated at the transcript level (Figure 1C and Appendix A), whereas TRAIL upregulation was only mild or not observed upon TNFα stimulation in the different cancer cell lines, which served as negative control. Interestingly, in certain breast cancer cell lines (i.e., MCF7 and HCC1954) and in the lung adenocarcinoma cell line A549, the upregulation of TRAIL was substantially high, ranging from 50-fold to greater than 100-fold higher than that in other breast cancer lines (e.g., in BT549, it was 8-fold higher) and other cancer lines (e.g., in liver hepatocellular carcinoma, HepG2, it was 34-fold higher). Generally, the levels of TRAIL protein expression followed the pattern of transcript expression. For instance, the immunofluorescence analysis revealed a greater increase in TRAIL protein in MCF7 and A549 cells than in BT549 cells (Figure 1D). Due to this very dynamic and significant upregulation of TRAIL mRNA and, consequently, protein expression by IFNα stimulation, we decided to investigate the mechanism of its upregulation at the transcriptional level.

### 3.2. IFNα-Induced Upregulation of TRAIL Is Linked to Chromatin Accessibility, Thus Enhancer Clusters, and Their Disruption Decreases TRAIL Expression

To understand the dynamic upregulation of TRAIL by IFNα in the different cancer types, we examined the genomic regulatory landscape to unravel the gene regulatory architecture (i.e., accessible chromatin carrying either disperse enhancers or enhancer clusters). We hypothesized that enhancer clusters could be associated with the upregulation of TRAIL mRNA in the cancer cells in which we observed higher levels of TRAIL mRNA expression induced by IFNα. To identify regulatory elements upstream and downstream of the TRAIL coding region, we used publicly available H3K27ac ChIP-seq data from human cancer cell lines, together with DNase I hypersensitivity data from a panel of cell lines, available through ENCODE. Both assays are widely used to identify accessible DNA regions that commonly carry regulatory elements [16,32,33]. From this analysis, we located two enhancer clusters, upstream and downstream of the TRAIL coding region, containing several putative regulatory elements (Figure 2A). Interestingly, we found that in the cell lines MCF7 and A549, in which we observed that upon IFNα stimulation, TRAIL transcript levels increased approximately 50-fold and 100-fold, respectively, TRAIL was associated with either upstream or both (upstream and downstream) identified enhancer clusters. On the contrary, we did not observe this association in cancer cells such as BT549 (Figure 2A), where TRAIL mRNA levels were only increased 8-fold upon IFNα stimulation. Together, this suggests that enhancer clusters, thus highly open chromatin regions, could be responsible for increasing TRAIL expression to high levels by IFNα.

Next, we wondered whether the correlation between TRAIL gene expression levels and the degree of chromatin “openness” observed in cancer cell lines is also a phenotype of patient tumors. To answer this question, we used publicly available ATAC-seq and RNA-seq data from 404 pan-cancer tumors deposited in TCGA. We focused our analysis on eight distal regulatory DNA regions (regulatory elements 1 to 8), upstream and downstream of the TRAIL coding region, identified through DNase I hypersensitivity data from ENCODE (Figure 2A). ATAC-seq information for regions within these distal enhancers was available for four of the eight regulatory elements (regulatory elements 2, 4, 7 and 8; Appendix A), indicating that these DNA regions are also open and accessible in patient tumors. Importantly, ATAC-seq chromatin accessibility for regulatory elements 2, 4 and 7 strongly and positively correlated with TRAIL RNA levels from pan-cancer tumors (Pearson coefficients = 0.4438, 0.5131 and 0.5068, respectively; *p* < 0.0001), while regulatory element 8 showed a smaller but significant positive correlation (Pearson coefficient = 0.2380; *p* < 0.0001) (Figure 2B). In summary, our findings show that chromatin “openness”—and cluster enhancers located in the most open regions—positively correlates with TRAIL mRNA expression. Moreover, our computational analyses in pan-cancer tumors suggest that a similar trend occurs in tumors derived from patients.

To confirm the regulatory function of enhancer clusters on TRAIL gene expression, we perturbed BRD4 binding to acetylated regions by treating cancer cell lines with BET inhibitors. Since enhancer clusters are sensitive to BET inhibitors, these inhibitors are a widely used tool to study whether enhancer clusters regulate the expression of associated genes. We found that BET inhibitors JQ1 and I-BET151 reduced TRAIL expression at the transcript (Figure 2C) and protein levels (Appendix A) in MCF7 and A549 cells, in which the TRAIL coding region is associated with enhancer clusters, but not in the breast cancer cell line BT549, which lacks an enhancer cluster association with TRAIL. Moreover, we observed in MCF7 and A549 cells that pretreatment with BET inhibitors 3 h prior to IFNα stimulation dramatically decreased TRAIL mRNA upregulation by IFNα at 6 h, an effect we did not observe in BT549 cancer cells, which lack an enhancer cluster for TRAIL (Figure 2D), suggesting that a functional enhancer cluster is necessary for the increase in TRAIL gene expression driven by IFNα. Finally, we confirmed that the reduction in TRAIL expression after BET inhibition was not due to cell death, as JQ1 or I-BET151 treatment did not affect cell survival (Appendix A).

### 3.3. IFNα-Induced TRAIL Accumulates Inside Epithelial Cancer Cells and Is Not Detected Extracellularly

Next, we asked whether TRAIL protein secretion augmented in the cell lines in which TRAIL expression was increased at the mRNA level following IFNα stimulation. Unexpectedly, we found that in contrast to THP-1 cells, in which soluble TRAIL protein is readily detectable in their supernatant upon IFNα stimulation (Figure 3A), soluble TRAIL was not detected in the supernatants of IFNα-stimulated MCF7, BT549 and A549 cells (data not shown). In addition, TRAIL protein expression on the surface of A549 or MCF7 cells was also not increased upon IFNα treatment (Figure 3B), unlike what we observed in THP-1 (Figure 1B). However, in agreement with our immunofluorescence results on epithelial cancer cells (Figure 1D), an increase in total TRAIL protein upon IFNα stimulation was observed by means of flow cytometry in permeabilized A549 or MCF7 cells, as well as in THP-1 cells (Figure 3C). Protein accumulation of IFNα-induced TRAIL could be mainly detected in the nucleus in the three epithelial cancer cells studied, as well as in several other subcellular compartments that vary across the analyzed cell lines (Figure 3D). For example, in the breast cancer cell line MCF7, a large percentage of cells showed a punctuated staining of TRAIL in the cytoplasm, which was compatible with vesicle localization. In subsets of A549 cells, TRAIL staining seemed to localize in a bigger compartment located next to the cell nucleus, which resembled the Golgi location, while in other A549 subsets, a punctuated cytoplasmic staining was observed. In BT549, TRAIL staining seemed to be mainly nuclear, with a small percentage of cells showing punctuated vesicle-like cytoplasmic staining or “Golgi”-like staining. Overall, our results show that IFNα-upregulated TRAIL protein accumulates intracellularly and is not secreted in MCF7, BT549 or A549 cancer cells, and since this accumulation was also observed in BT549 cells (where TRAIL is not regulated by cluster enhancers), our results show that intracellular TRAIL protein accumulation is independent of TRAIL transcript upregulation by enhancer clusters.

### 3.4. IFNα-Induced Internal TRAIL Accumulation Does Not Activate Apoptosis of Cancer Cells

To test whether IFNα-induced intracellular TRAIL increases apoptosis, just as it is observed when cancer cells are exposed to exogenous recombinant soluble TRAIL, we performed Annexin V staining after treating breast and lung cancer cell lines exogenously with either IFNα or TRAIL. Our results showed that apoptosis only occurred when treating the cells with exogenous TRAIL (except for the MCF7 cell line, which has DNA copy number deletions in TRAIL receptors coding genes; Appendix A) and not when treating them with IFNα (Figure 4A and Appendix A). Similar results were obtained when using the A549 cancer line to monitor cell death by measuring PI uptake in real time with or without pan-caspase inhibitors after exogenous IFNα or TRAIL treatment. Here, we only observed caspase-mediated cell death upon treatment with exogenous TRAIL (Appendix A). Since both of these experiments demonstrate that IFNα-induced intracellular TRAIL does not increase apoptosis, next, we assayed cytotoxicity by measuring the accumulation of the cytosolic enzyme lactate dehydrogenase, which is released into the cell culture medium upon damage of the cell membrane during cell death. Treatment with IFNα, when compared to exogenous TRAIL, did not show a cytotoxic effect in most epithelial cancer cells studied, confirming that IFNα-induced intracellular TRAIL does not induce any type of cell death (Figure 4B), an effect that is independent of TRAIL association with enhancer clusters (Figure 2A).

Finally, we performed conditioned medium experiments to test whether IFNα-driven TRAIL upregulation can induce the death of epithelial cancer cells in a paracrine manner. To test this, we collected supernatants from IFNα-stimulated A549 (TRAIL-non-secreting) or THP-1 (TRAIL-secreting; Figure 3A) cells and incubated the TRAIL-sensitive cell line HeLa, employed as target cells [34], with either of these supernatants. Supernatants from both A549 and THP-1 were supplemented with BV6, a SMAC mimetic widely used to sensitize cells to death ligand-mediated cell death [35]. As expected, only supernatants collected from IFNα-stimulated THP-1 but not from IFNα-stimulated A549 cells induced the death of HeLa cells, showing that A549 cells do not secrete biologically active TRAIL upon IFNα stimulation (Figure 4C). Collectively, our results show that the IFNα-upregulation of TRAIL in cancer cells is not capable of inducing cell death, including in cells that are highly sensitive to apoptosis induction by exogenous TRAIL.

## 4. Discussion

Our work shows that TRAIL expression is upregulated by open and highly accessible enhancer clusters in cancer cells and that the levels of expression of TRAIL mRNA are likewise linked to the openness of distal cis-elements in pan-cancer patients. Moreover, we found that enhancer clusters dramatically enhance TRAIL upregulation by IFNα in breast and lung cancer cells but that the consequent increase in TRAIL expression at the protein level does not result in an increase in extracellular TRAIL, neither in the cancer cell supernatants nor on their surface. Contrary to what we and others have observed in immune cells in which IFN stimulation leads to higher levels of transmembrane and soluble TRAIL protein [36,37,38], in the epithelial cancer cell lines analyzed, we observed that TRAIL protein accumulated intracellularly upon IFNα stimulation. This intracellular TRAIL protein is not capable of inducing apoptosis in cancer cells in an auto- or paracrine manner, a finding which contrasts with what has been previously reported by others for other cancer types [8,10]. Furthermore, in the epithelial cancer cell lines studied, we found that IFNα-induced TRAIL accumulates intracellularly in different cell compartments, appearing to localize, in some instances, in vesicles. Previous studies in colorectal cancer cells have shown that TRAIL is constitutively secreted in extracellular vesicles. These studies have also shown that the pro-apoptotic function of vesicle-associated TRAIL can be blocked by anti-TRAIL antibodies, indicating that TRAIL is on the surface of vesicles [39]. Since our ELISA experiments did not detect TRAIL either as a soluble protein or bound to vesicles upon the IFNα stimulation of the epithelial cancer cells analyzed and since the supernatant of A549 cells failed to induce the cell death of sensitized target cells (Figure 4C), we conclude that TRAIL is not secreted in any form by the cancer cells analyzed in our study. Intracellular TRAIL protein accumulation in epithelial cancer cells could be a consequence of either deficient post-translational TRAIL modifications or the absence of trafficking proteins in cancer cells, both of which could prevent the correct translocation of TRAIL to the cell membrane. It is also possible that IFNα could be driving counteracting mechanisms in cancer cells that retain TRAIL intracellularly in these cells, whereas in immune cells, these mechanisms would not be operative.

On the other hand, clustered enhancers spanning across long regions of DNA have been described as critical cis-regulatory elements for the regulation of oncogenes [40]. Because of the rapid advancement in DNA sequencing technologies, numerous tumor-promoting genes associated with these regulatory regions have recently been found in a wide variety of cancer types [41]. So far, enhancer clusters have been linked to the enhanced expression of their associated genes. In cancer, the epigenetic upregulation of oncogenes is an important mechanism that contributes to tumor evolution during malignant progression. Thus, we speculate that in malignant cells in which TRAIL expression is epigenetically enhanced, the role of TRAIL could differ from its bullet-type killing function extensively reported for transmembrane or secreted TRAIL in immune cells. In cancer, apart from triggering apoptosis, TRAIL binding to TRAIL receptors activates non-cell death signaling, which results in the activation of NF-κB and other pro-tumorigenic pathways [42,43,44,45,46]. However, most non-canonical TRAIL signaling also requires TRAIL binding to extracellular domains of TRAIL receptors.

Since we observed that (1) TRAIL protein can accumulate intracellularly, and in the nucleus, upon IFNα stimulation of the epithelial cancer cells we studied and (2) TRAIL can be upregulated in cancer due to epigenetic mechanisms usually found to be associated with oncogenes, a provocative possibility is that IFNα-upregulated TRAIL works as a pro-tumoral molecule in a group of epithelial cancer cells. To our knowledge, the role of intracellular TRAIL has not been reported, and in fact, it could have a novel internal role that is different from apoptosis induction, as it has been shown for other cytokines [47,48]. Thus, whether intracellular TRAIL can be rendered functional as an apoptosis inducer under certain circumstances and whether it has a different role when induced by type I IFNs in cancer cells are intriguing questions that remain to be addressed in future studies.

The fact that exogenously added TRAIL specifically induces cancer cell death with low toxicity to normal cells [49,50] led to the development of anticancer therapies modulating the interaction between TRAIL protein and TRAIL receptors to selectively kill cancer cells. One of these therapies consisted in the development of TRAIL receptor agonists [51,52], which, contrary to expectations, showed limited efficacy in cancer patients [53,54]. Explaining this low efficacy, it was found that the expression of TRAIL receptors (TRAIL-R1 and TRAIL-R2) is dynamic across cancers [55,56]. We observed that treatment with recombinant soluble TRAIL induced apoptosis in the majority of cancer cell lines tested. However, the ability of TRAIL to induce cell death greatly varied across cell lines. This different sensitivity of the studied cell lines to exogenous TRAIL can be explained at least in part by genetic alterations in TRAIL receptor genes (Appendix A). Thus, TRAIL receptor status could influence the response of tumor cells to TRAIL receptor agonists. In this study, we also postulate that TRAIL intracellular accumulation could be another factor contributing to cancer cell apoptosis impairment when using IFNα to stimulate its production in cancer cells. Thus, for the reasons presented here, current efforts focus on identifying new therapeutic targets with the goal of increasing TRAIL-mediated cytotoxicity [57,58,59,60].

## 5. Conclusions

In conclusion, we report that IFNα-induced overexpression of TRAIL in breast and lung cancer cells is driven by enhancer clusters. Yet, no extracellular TRAIL protein production nor higher TRAIL-induced apoptosis was achieved in an autocrine or paracrine manner by means of IFNα stimulation in the epithelial cancer cells studied. On the contrary, this IFNα-upregulated TRAIL protein accumulated intracellularly. Further studies are required to clarify any additional role played by the intracellular TRAIL protein in cancer cells.

## Figures and Tables

**Figure 1 cancers-15-00967-f001:**
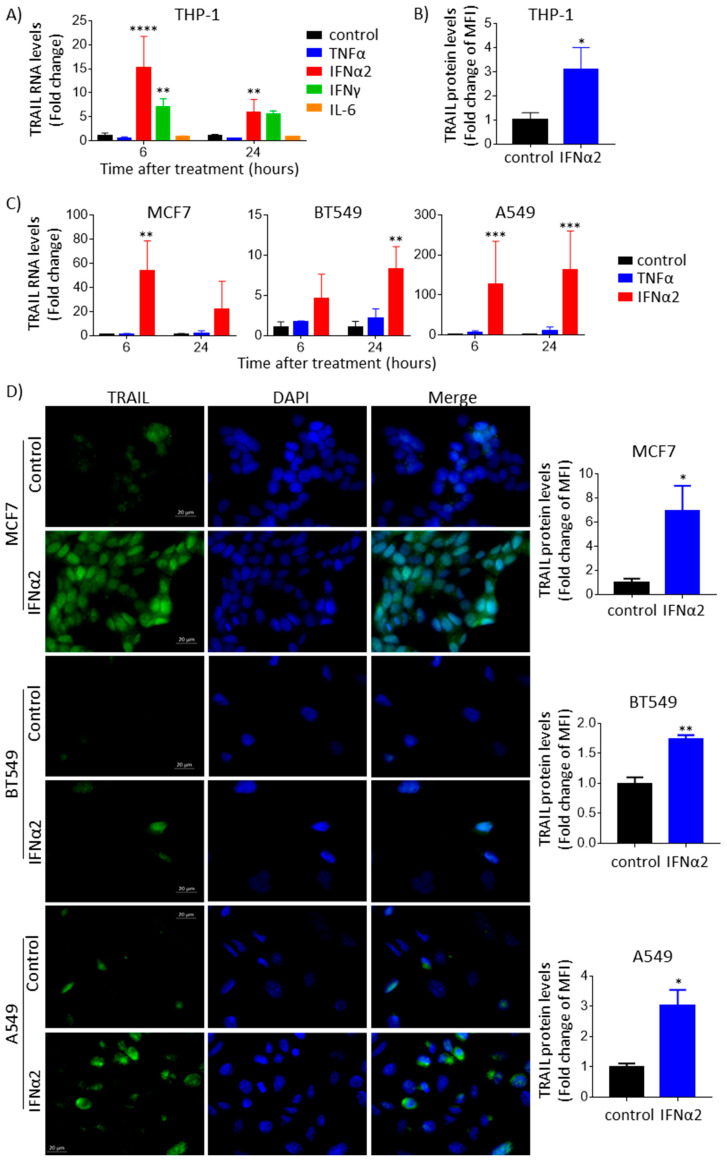
TRAIL RNA and protein levels are upregulated by IFNα in cancer cells. (**A**) TRAIL RNA expression increased upon IFNα and IFNγ treatment but not upon TNFα or IL-6 treatment in THP-1. Cells were treated with 1 μg/mL TNFα, 0.8 μg/mL IFNα2, 1 μg/mL IFNγ or 100 ng/mL IL-6 for 6 or 24 h, and TRAIL levels were quantified using qPCR. ANOVA followed by Dunnett’s multiple comparison test was performed (** *p* < 0.01 and **** *p* < 0.0001). (**B**) TRAIL protein levels on the cell surface increased upon IFNα treatment in THP-1. Cells were treated with 0.8 μg/mL IFNα2 for 24 h, and membrane TRAIL protein was quantified in fresh cells using flow cytometry. Two-tailed *t*-test was performed (* *p* < 0.05). (**C**) TRAIL RNA expression increased upon IFNα treatment but not upon TNFα treatment in cancer cell lines. Breast cancer cell lines (MCF7 and BT549) and lung cancer cells (A549) were treated with 1 μg/mL TNFα or 0.8 μg/mL IFNα2 for 6 or 24 h, and TRAIL levels were quantified using qPCR. ANOVA followed by Dunnett’s multiple comparison test was performed (** *p* < 0.01 and *** *p* < 0.001). (**D**) TRAIL protein levels increased upon IFNα treatment in cancer cell lines. Cells were treated with 0.8 μg/mL IFNα2 for 24 h (A549) or 48 h (MCF7 and BT549), and total TRAIL protein was quantified using immunofluorescence. TRAIL signal is shown in green. Nuclei were stained with DAPI (blue). Representative images are shown. Two-tailed *t*-test was performed (* *p* < 0.05 and ** *p* < 0.01). MFI: mean fluorescence intensity.

**Figure 2 cancers-15-00967-f002:**
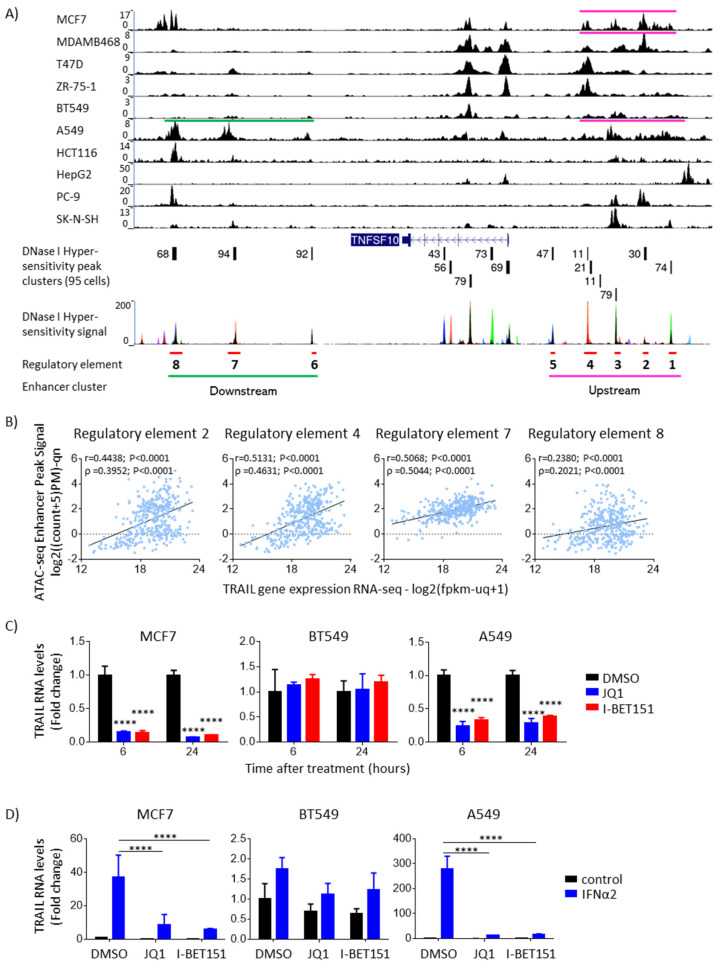
Enhancer clusters mediate TRAIL upregulation by IFNα. (**A**) TRAIL is associated with enhancer clusters in specific cancer cell lines. Publicly available H3K27ac ChIP-seq data for several cell lines from breast cancer (MCF7, MDAMB468, T47D, ZR-75-1 and BT549) and other solid cancers (A549, HCT116, HepG2, PC-9 and SK-N-SH) were analyzed. Peaks of H3K27ac signal represent accessible chromatin and were found upstream and downstream of TRAIL (TNFSF10), which is transcribed from the opposite strand (right to left). Distal H3K27ac peaks overlapping with DNase I hypersensitive sites (a mark of open regulatory DNA) indicate accessible open DNA regions carrying putative regulatory elements. Distal DNase I hypersensitive sites, called regulatory elements 1 to 8 and shown in red in the diagram, were identified through DNase I hypersensitivity assays in 95 cell lines from ENCODE. Only distal DNase I hypersensitive regions that were found in more than 20 out of the 95 cell lines were considered as putative regulatory elements in this analysis. Regulatory elements 1 to 5 cluster together upstream of TRAIL (to the right in the figure), whereas regulatory elements 6 to 8 cluster together downstream of TRAIL (to the left in the figure). Upstream and downstream enhancer clusters are represented as pink and green lines, respectively. As observed in the H3K27ac data, MCF7, MDAMB468 and A549 cell lines show an upstream enhancer cluster, while only A549 shows a downstream enhancer cluster. (**B**) Chromatin accessibility at distal enhancer regions upstream and downstream of TRAIL-coding DNA correlates with TRAIL RNA levels in pan-cancer tumors. Scatter plots show correlation between TRAIL RNA-seq expression and ATAC-seq data on chromatin accessibility for 4 open DNA regions (regulatory elements 2, 4, 7 and 8) from 404 pan-cancer tumors from TCGA dataset. Pearson (r) and Spearman (ρ) correlation coefficients and *p*-values are shown for each plot. (**C**) Blocking BRD4 binding to highly acetylated enhancer regions with BET inhibitors reduced TRAIL RNA expression in MCF7 and A549, but not in BT549. Cells were treated with vehicle (DMSO), 1 μM JQ1 or 1 μM I-BET151 for 6 h. After treatment, changes in TRAIL RNA levels were analyzed using qPCR. ANOVA followed by Dunnett’s multiple comparison test was performed (**** *p* < 0.0001). (**D**) Enhancer clusters enhanced TRAIL upregulation by IFNα. Cells were pre-treated with vehicle (DMSO), 1 μM JQ1 or 1 μM I-BET151 for 3 h, followed by stimulation with 0.8 μg/mL IFNα2 for 6 h. After treatment, changes in TRAIL RNA levels were analyzed using qPCR. ANOVA followed by Dunnett’s multiple comparison test was performed (**** *p* < 0.0001).

**Figure 3 cancers-15-00967-f003:**
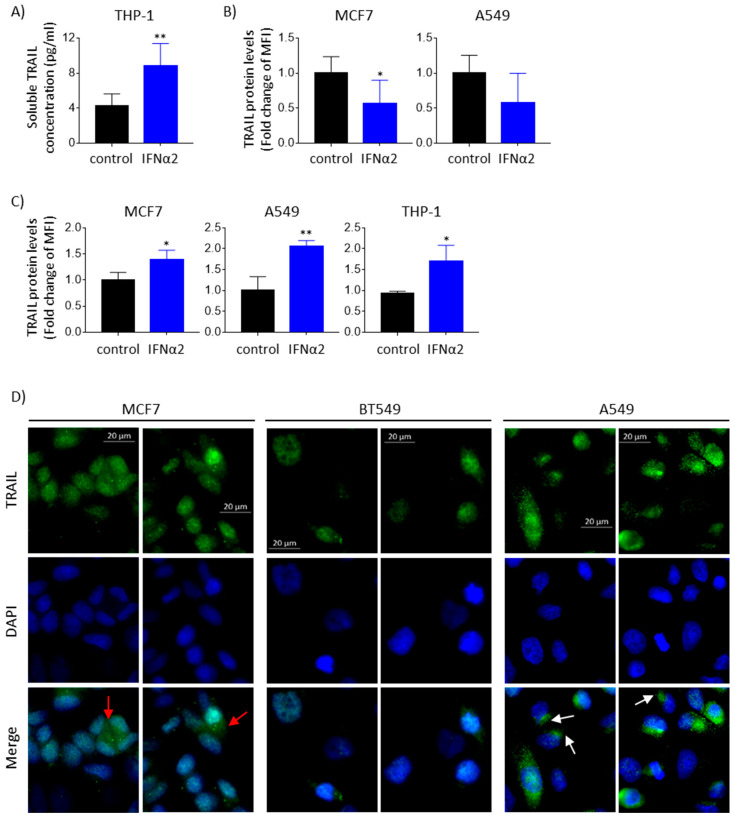
IFNα-induced TRAIL protein accumulates intracellularly in epithelial cancer cells. (**A**) Soluble TRAIL protein levels increased upon IFNα stimulation in THP-1. Cells were treated with 0.8 μg/mL IFNα2 for 24 h, and TRAIL protein in the culture media was quantified using ELISA. Two-tailed *t*-test was performed (** *p* < 0.01). (**B**) TRAIL protein levels on the cell surface did not increase upon IFNα stimulation in epithelial cancer cells. MCF7 and A549 were treated with 0.8 μg/mL IFNα2 for 48 h, and membrane TRAIL protein was quantified in fresh cells using flow cytometry. Two-tailed *t*-test was performed (* *p* < 0.05). (**C**) Total TRAIL protein levels increased upon IFNα stimulation in epithelial and immune cancer cells. MCF7, A549 and THP-1 were treated with 0.8 μg/mL IFNα2 for 48 h, and total TRAIL protein was quantified in fixed and permeabilized cells using flow cytometry. Two-tailed *t*-test was performed (* *p* < 0.05 and ** *p* < 0.01). (**D**) IFNα-induced TRAIL was detected intracellularly in epithelial cancer cells. Cells were treated with 0.8 μg/mL IFNα2 for 24 h (A549) or 48 h (MCF7 and BT549), and total TRAIL protein was quantified using immunofluorescence. TRAIL signal is shown in green. Nuclei were stained with DAPI (blue). Red arrows mark punctuated cytoplasmic TRAIL staining. White arrows mark TRAIL staining in a big compartment next to the cell nucleus. Representative immunofluorescence images are shown. MFI: mean fluorescence intensity.

**Figure 4 cancers-15-00967-f004:**
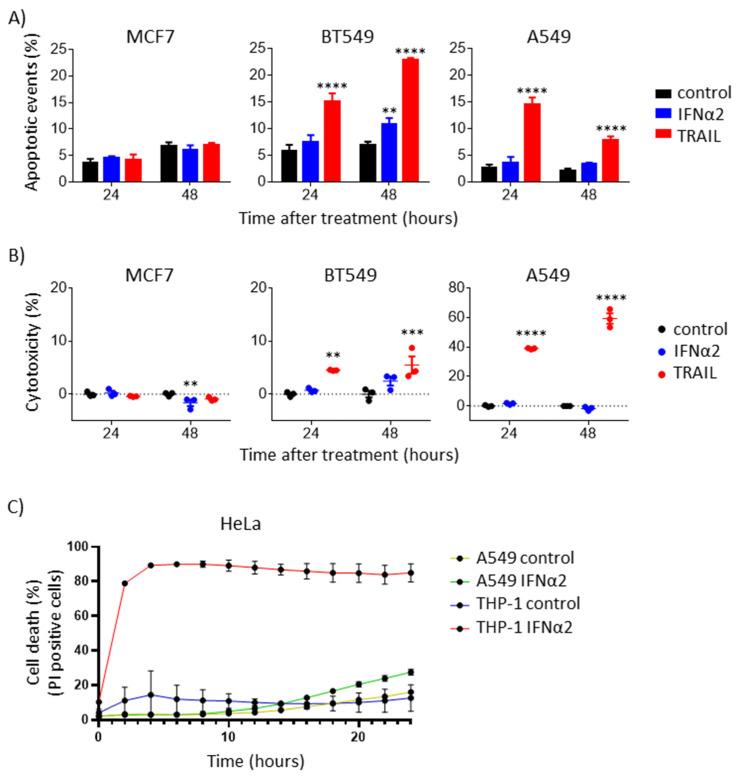
TRAIL upregulation by IFNα in epithelial cancer cells is not enough to induce apoptosis. (**A**) TRAIL upregulation by IFNα did not induce apoptosis in most cancer cells. Breast cancer cells MCF7 and BT549 and lung cancer cells A549 were treated with 0.8 μg/mL IFNα2 or 0.5 μg/mL TRAIL. After 24 or 48 h, apoptosis was quantified with Annexin V staining using flow cytometry. ANOVA followed by Dunnett’s multiple comparison test was performed (** *p* < 0.01 and **** *p* < 0.0001). (**B**) TRAIL upregulation by IFNα did not induce cytotoxicity in cancer cells. Breast cancer cells MCF7 and BT549 and lung cancer cells A549 were treated with 0.8 μg/mL IFNα2 or 0.5 μg/mL TRAIL. After 24 or 48 h, cytotoxicity was quantified using the lactate dehydrogenase cytotoxicity assay. ANOVA followed by Dunnett’s multiple comparison test was performed (** *p* < 0.01, *** *p* < 0.001 and **** *p* < 0.0001). (**C**) TRAIL upregulation by IFNα in epithelial cancer cells did not induce cell death in a paracrine manner. THP-1 and A549 were treated with 0.8 μg/mL IFNα2 or control. After 48 h, the supernatants were collected, centrifuged and supplemented with 1 μM BV6 and 1 μg/mL PI. Cell death induction by the conditioned media was monitored in real time for 24 h using HeLa as target cells. Cell death is represented as the percentage of PI-positive cells.

## Data Availability

H3K27ac ChIP-seq data analyzed in this study are publicly available from ENCODE at “https://www.encodeproject.org/” (accessed on 15 January 2022) [23] or from GEO at “https://www.ncbi.nlm.nih.gov/geo/” (accessed on 15 January 2022) [61]. The accession numbers are listed in Appendix A. Data were visualized using UCSC Genome Browser [24]. Publicly available ATAC-seq and RNA-seq TCGA data were downloaded from UCSC XenaBrowser Platform at “https://xenabrowser.net/” (accessed on 30 January 2022) [26]. The Cell Line Encyclopedia data used in this study [27] were analyzed using cBioPortal at “https://www.cbioportal.org/” (accessed on 21 June 2022) [28].

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
