# Peer review of "Enhancer Clusters Drive Type I Interferon-Induced TRAIL Overexpression in Cancer, and Its Intracellular Protein Accumulation Fails to Induce Apoptosis"

_cancers, 2023, doi:10.3390/cancers15030967_

Round 1
Reviewer 1 Report
The manuscript by Di Benedetto et al. shows a work well developed, with the adequate methodology, and with relevant observations for cancer biology.
The main concern comes from the absence of intracellular localization of the type I IFN-induced TRAIL. Previous reports indicated that TRAIL could be stored in multivesicular bodies in certain solid tumor cells, such as colorectal cancer cells (Huber et al., Gastroenterelogy, 128:1796; 2005) or melanoma cells (Martinez-Lorenzo et al., Exp. Cell Res. 295:315, 2004), and be secreted to the supernatant associated with extracellular vesicles (EV), maintaining in this way its pro-apoptotic potential. In the Huber study the secretion of TRAIL associated with EV seemed to be constitutive, while in the Martinez study it was induced by the lectin PHA or by the more specific melanocyte-stimulating hormone (MSH).
Authors should study by confocal microscopy the intracellular compartment in which TRAL is stored in the type I IFN-treated tumors and/or the type of additional stimulation that could induce secretion of soluble TRAIL or of EV-associated TRAIL, resulting or not in tumor cell death.
These additional data will very much enrich the conclusions of the manuscript.
Minor point: The inhibition of TRAIL-induced A549 cell death by Z-VAD-fmk should be shown at least in supplemental material.
Author Response
The manuscript by Di Benedetto et al. shows a work well developed, with the adequate methodology, and with relevant observations for cancer biology.
The main concern comes from the absence of intracellular localization of the type I IFN-induced TRAIL. Previous reports indicated that TRAIL could be stored in multivesicular bodies in certain solid tumor cells, such as colorectal cancer cells (Huber et al., Gastroenterelogy, 128:1796; 2005) or melanoma cells (Martinez-Lorenzo et al., Exp. Cell Res. 295:315, 2004), and be secreted to the supernatant associated with extracellular vesicles (EV), maintaining in this way its pro-apoptotic potential. In the Huber study the secretion of TRAIL associated with EV seemed to be constitutive, while in the Martinez study it was induced by the lectin PHA or by the more specific melanocyte-stimulating hormone (MSH).
We thank the reviewer for raising this point. In our manuscript, we showed in Fig 4C that IFNα-induced TRAIL is not released to the supernatant, either as a soluble protein or in EV (as observed in the colorectal cancer study by Huber et al (1), where released EV membrane-bound TRAIL can induce TRAIL receptor-mediated apoptosis of target cells). In our experiments, the supernatant collected from IFNα-stimulated A549 cells did not induce TRAIL receptor-mediated cell death of sensitized HeLa cells (Fig 4C). Furthermore, the fact that the apoptotic activity of EV-associated TRAIL is blocked using an anti-TRAIL antibody, as shown in the colorectal cancer study (1), indicates that TRAIL is exposed on the EV membrane and, thus, ELISA assays should detect EV-bound TRAIL. Yet, we did not detect TRAIL by ELISA in IFNα-stimulated epithelial cancer cells, confirming that IFNα-induced TRAIL is not secreted in EV in epithelial cancer cells. In sum, we are thankful to the Reviewer for bringing up EV-associated TRAIL. We realize that this is an important point that should be mentioned in our work. As a result, we now mention the possibility of cancer cells releasing EV-bound TRAIL and cite the corresponding publication in the “Discussion” section (lines 470 to 479), and we also discuss that upon IFNα stimulation of epithelial cancer cells EV-bound TRAIL is not detected in our study.
Regarding the search of additional stimulation, apart from IFNα, that could induce secretion of soluble TRAIL or of EV-associated TRAIL, this is a very interesting question that we will consider for future studies. However, we did not proceed with further analyses on this point because this topic is outside the scope of our manuscript, as our main focused is that of TRAIL upregulation by IFN.
Authors should study by confocal microscopy the intracellular compartment in which TRAL is stored in the type I IFN-treated tumors and/or the type of additional stimulation that could induce secretion of soluble TRAIL or of EV-associated TRAIL, resulting or not in tumor cell death.
These additional data will very much enrich the conclusions of the manuscript.
We thank the Reviewer for bringing up this important question concerning the intracellular compartment in which IFNα-induced TRAIL is stored. We agree with the reviewer that confocal microscopy studies using organelle markers would be very helpful to precisely determine the localization of IFNα-induced TRAIL. However, the fact that we were only given 10 days for revising this manuscript makes it impossible for us to perform such studies. Thanks to the reviewer’s comment, we realized that our immunofluorescence images were too small for visualizing TRAIL distribution in the cell and now we include zoomed-in images in Fig 3D and a detailed description (lines 381 to 394). A closer look at the enlarged images suggests that protein accumulation of IFNα-induced TRAIL could be detected mainly in the nucleus in the three epithelial cancer cells studied, as well as in other several subcellular compartments that vary across the analyzed cell lines. For example, in the breast cancer cell line MCF7, a large percentage of cells show a punctuated staining of TRAIL in the cytoplasm, which is compatible with vesicle localization. In subsets of A549 cells, TRAIL staining seems to localize in a bigger compartment located next to the cell nucleus, which resembles Golgi location, while in other A549 subsets a punctuated cytoplasmic staining is observed. In BT549, TRAIL staining seems to be mainly nuclear, with a small percentage of cells showing punctuated vesicle-like cytoplasmic staining or “Golgi”-like staining. Thanks to the reviewer comment, we found that IFNα-induced TRAIL could be retained intracellularly in different cell compartments including vesicles in epithelial cancer cell lines.
Minor point: The inhibition of TRAIL-induced A549 cell death by Z-VAD-fmk should be shown at least in supplemental material.
We apologize for not being clear and omitting mentioning that the inhibition of TRAIL-induced A549 cell death by Z-VAD-FMK was in supplemental material (Fig S3C). However, thanks to the reviewer’s concern we made this clearer in the manuscript, by rewording the “Results” section (lines 419 to 422), and by describing the Z-VAD-FMK treatment in the “Real-time cell death assay” section in the “Methods” to clarify this experiment and interpretation.
In addition to the changes made in response to the Reviewer’s comments, we made some minor grammatical changes throughout the manuscript that improve its reading. These minor changes are highlighted in yellow.
References:
- Huber V, Fais S, Iero M, Lugini L, Canese P, Squarcina P, et al. Human colorectal cancer cells induce T-cell death through release of proapoptotic microvesicles: role in immune escape. Gastroenterology. 2005 Jun;128(7):1796–804.
Reviewer 2 Report
In this article Benedetto et al found IFNα-induced upregulation of TRAIL in cancer cells is driven by enhancer clusters. Yet, no extracellular TRAIL protein production nor higher TRAIL-induced apoptosis is achieved by IFNα stimulation in the epithelial cancer cells. Further studies are needed to clarify the additional function of intracellular TRAIL protein in cancer cells.
This study raised a question, rather than solved this question. The key issue is the mechanism why accumulation of intracellular TRAIL did not induce cancer cell apoptosis. The authors just mentioned these intriguing questions that remain to be addressed in future studies. Since in vitro cancer killing effects are not verified, the scientific value of this study is restricted. I would like to reject this manuscript.
Author Response
In this article Benedetto et al found IFNα-induced upregulation of TRAIL in cancer cells is driven by enhancer clusters. Yet, no extracellular TRAIL protein production nor higher TRAIL-induced apoptosis is achieved by IFNα stimulation in the epithelial cancer cells. Further studies are needed to clarify the additional function of intracellular TRAIL protein in cancer cells.
This study raised a question, rather than solved this question. The key issue is the mechanism why accumulation of intracellular TRAIL did not induce cancer cell apoptosis. The authors just mentioned these intriguing questions that remain to be addressed in future studies. Since in vitro cancer killing effects are not verified, the scientific value of this study is restricted. I would like to reject this manuscript.
With all due respect to the Reviewer, we are confident that our work has extensive scientific value and provides novel information to the cancer field and TRAIL biology. Through our work we:
- Identified a novel mechanism of gene regulation involving cluster of enhancers that has not been previously described to explain the IFNα-driven overexpression of TRAIL in cancers. For this, we combined computational analyses of TRAIL genomic regions obtained from cancer cell lines and from breast cancer patients’ repositories and performed in-vitro drug inhibition of enhancer clusters to demonstrate their regulatory function on TRAIL gene expression.
- Described and demonstrated for the first time that increased IFNα-induced TRAIL is accumulated in certain cancer cells and it is not secreted, thus, contradicting a role that has been extensively studied for TRAIL in immune and certain cancer cells. For this we performed immunofluorescence, flow cytometry, and ELISA in-vitro experiments.
- Reported and demonstrated that IFNα-induced intercellular TRAIL does not induce apoptosis or any type of cell death of cancer cells in an autocrine or paracrine manner. We showed this by measuring cell death (LDH release, PI staining) and apoptosis (Annexin V staining, PI staining using caspase inhibitors). Moreover, and contrary to what the Reviewer notes, we did perform in-vitro killing assays (Fig 4C) to verify that the IFNα-induced TRAIL was incapable of killing other cells in a paracrine manner as it has been reported for certain cancer cells.
To further emphasize and clarify the main points of our work, now we substantially modified the manuscript based on other Reviewers’ suggestions (see highlighted blue text in the manuscript). Revisions that are not highlighted in blue are:
1) moving our computational TRAIL enhancer cluster analysis figure from supplementary material to the main text (Fig 2A).
2) Removing speculative conclusion or future questions from the conclusion section.
Hopefully these new revisions to the manuscript clarify to the Reviewer that our main point was not to find a mechanism explaining why accumulation of intracellular TRAIL did not induce cancer cell apoptosis, but rather find a mechanism that explains TRAIL overexpression driven by IFN in cancer cells, a topic that goes in line with the “Alterations in Cis-Regulatory Elements in Human Cancer” special edition of the journal “CANCERS”
We thank the Reviewer for the time committed to reading and reviewing our work and we kindly ask for reconsideration of our manuscript.
Reviewer 3 Report
The authors investigated the molecular mechanisms of how TRAIL expression was upregulated under INF-stimulation. They showed that the consequence of the over expression was intracellular accumulation but incapable of inducing apoptosis. The finding is overall interesting but some improvement could be made.
1. The authors could show ChIP-seq (Mediator or H3K27ac) comparison between IFN-stimulated cells and control cells to demonstrate that these enhancer clusters are indeed activated by drug treatment.
2. The authors only used one marker to measure apoptotic event (Annexin V). More markers should be examined, particularly the caspase activity.
3. In Figure 3D, the authors aimed to show the cytosolic accumulation of TRAIL protein but the image is too small. Can zoom-in image be placed here? This may also help identify which sub-cellular components contain the protein that is not able to be secreted. Maybe the authors should isolate subcellular components respectively by fractionation and detect the enrichment of the TRAIL protein.
4. Since CRISPR/Cas9 is nowadays available and easy to be used to edit the genome, the authors should respectively Knockout these enhancers to demonstrate the function of them in driving the expression of TRAIL mRNA under stimulation.
Round 2
Reviewer 1 Report
The authors have addressed the concerns suggested in the previous revision.
Reviewer 3 Report
The authors have adequately addressed my previous concerns.